**PLOS** NEGLECTED TROPICAL DISEASES

# *Clock* genes regulate mating activity rhythms in the vector mosquitoes, *Aedes albopictus* and *Culex quinquefasciatus*

**Shuang Liu**[1☉], **Jiayong Zhou**[1☉], **Ling Kong**[1☉], **Yiquan Cai**[1], **Hongkai Liu**[1], **Zhensheng Xie**[1], **Xiaolin Xiao**[1], **Anthony A. James**[2,3], **Xiao-Guang Chen**[1]*

**1** Department of Pathogen Biology, Institute of Tropical Medicine, School of Public Health, Southern Medical University, Guangzhou, China, **2** Department of Microbiology & Molecular Genetics, University of California, Irvine California, United States of America, **3** Department of Molecular Biology & Biochemistry, University of California, Irvine California, United States of America

☉ These authors contributed equally to this work.
* xgchen@smu.edu.cn.

**Data Availability Statement:** All relevant data are within the manuscript and its Supporting Information files.

## Abstract

### Background

Endogenous circadian rhythms result from genetically-encoded molecular clocks, whose components and downstream output factors cooperate to generate cyclic changes in activity. Mating is an important activity of mosquitoes, however, the key aspects of mating rhythm patterns and their regulatory mechanisms in two vector mosquito species, *Aedes albopictus* and *Culex quinquefasciatus*, remain unclear.

### Methodology/Principal findings

We determined and compared the diel mating activity rhythms of these two mosquito species and discovered that *Ae. albopictus* had mating peaks in the light/dark transition periods (ZT0-3 and ZT9-12), while *Cx. quinquefasciatus* only had a mating peak at ZT12-15. Knockouts of the *clock* (*clk*) orthologous genes (*Aalclk* and *Cxqclk*) resulted in phase delay or phase reversal of the mating peaks in *Ae. albopictus* and *Cx. quinquefasciatus*, respectively. In addition, the temporal expression pattern of the desaturase orthologous genes, *desat1*, in both mosquito species was also different in respective wild-type strains and showed phase changes similar to the mating rhythms in *clk* mutant strains. Inhibition of *desat1* expression resulted in decreased mating activity in male mosquitoes of both species but not females. In addition, *desat1* regulated cuticular hydrocarbons' synthesis in both species. Silencing *desat1* in male *Ae. albopictus* resulted in decreases of nonadecane and tricosane, which promoted mating, with concomitant increases of heptacosane, which inhibited mating. Silencing *desat1* in male *Cx. quinquefasciatus* also resulted in decreases of tricosane, which promoted mating.

### Conclusions/Significance

These results suggest that *Aalclk* and *Cxqclk* have significant roles in the mating activity rhythms in both *Ae. albopictus* and *Cx. quinquefasciatus* by regulating the temporal

**Funding:** This work was supported by grants from the National Key Research and Development Program of China (2020YFC1200100), the National Natural Science Foundation of China (81829004, 31830087), and the National Institutes of Health, USA (AI136850) to X-G.C. The funders had no role in study design, data collection and analysis, decision to publish, or preparation of the manuscript.

**Competing interests:** The authors have declared that no competing interests exist.

expression of the *desat1* gene under LD cycles, which affects sex pheromone synthesis and mating. This work provides insights into the molecular regulatory mechanism of distinct mating rhythm of *Ae. albopictus* and *Cx. quinquefasciatus* and may provide a basis for the control of these two important vector mosquitoes.

## Author summary

*Aedes albopictus* and *Culex quinquefasciatus* are globally-distributed and important vector mosquito species that transmit a variety of pathogens. The two mosquito species have different circadian patterns, with *Ae. albopictus* being diurnal and *Cx. quinquefasciatus* being nocturnal. Mating is an important activity of mosquitoes, however, the mating rhythms and their regulatory mechanisms in *Ae. albopictus* and *Cx. quinquefasciatus* have yet to be described. Here, we compared the diel mating activities of these two mosquito species and discovered that *Ae. albopictus* and *Cx. quinquefasciatus* have distinct mating rhythms, which can be entrained by daily LD cycles. CRISPR/Cas9 technologies were used to knock out the core clock gene *clock* (*clk*) in the two mosquito species and found that *Aalclk* and *Cxqclk* have significant roles in the mating activity rhythms in both *Ae. albopictus* and *Cx. quinquefasciatus* by regulating the temporal expression of the desaturase *desat1* gene, which can affect sex pheromone synthesis and mating. These findings establish the basis for an in-depth understanding of the mating rhythms for mosquito vectors, which may provide insights in the development of novel and precise mosquito control.

## Introduction

Daily changes in light and temperature exert strong selective pressures on organisms and entrain them to distinct physiological and behavioral activity cycles with an ~24 hour rhythm [1–4]. It has been confirmed in *Drosophila melanogaster* and mammals that these rhythms are regulated and maintained by molecular circadian clocks [5]. In *D. melanogaster*, the expression products of core clock genes *clock* (*clk*) and *cycle* (*cyc*) act as transcriptional factors that heterodimerize and bind to E-boxes of other core clock genes *period* (*per*) and *timeless* (*tim*) to activate their transcription [6,7]. *per* and *tim* mRNAs are translated and their products then repress *clk*-mediated transcription [8]. Such transcription-translation feedback loop (TTFL) comprised of these core clock genes is the key molecular basis of the circadian clock. Although some components differ in function, the TTFL is conserved from insects to mammals. These genetically-encoded circadian clocks can respond to environmental cues (termed zeitgeber) such as light and control the expression of downstream output genes throughout the body, thereby temporally controlling the behavior and physiology to accommodate and exploit daily recurring environmental changes [5,9].

Mating behavior in animals is important and fundamental for species reproduction. It can be a complex process that depends on the accuracy of temporal and spatial coordination of the behaviors of both males and females. This coordination is critical, and short- (choosing the correct mate) and long-term (evolutionary) processes may be regulated by molecular circadian clocks [10]. Insects in particular show rhythmicity in mating behavior. For example, the mating activity of *D. mercatorum*, has diel rhythms under 12h light:12h dark (LD) cycles [11]. In LD cycles, the mating activity during the day is high in both *D. melanogaster* and *D. simulans* [12]. Moths (lepidoptera) mate mainly at night in order to elude predators [13,14]. The African malaria mosquito, *Anopheles gambiae*, mates at dusk during the first hour of the dark phase

[15]. However, the genetic and molecular bases for this behavior rhythm have yet to be described in many insect species.

*Aedes albopictus* is a diurnal mosquito that prefers to feed on humans and livestock in the early morning and evening. It is an important vector of dengue, Zika, and chikungunya viruses [16,17]. *Culex quinquefasciatus* is a nocturnal, anthropophilic mosquito that can transmit filariasis and Japanese encephalitis [18,19]. These mosquitoes are distributed globally and the pathogens they transmit represent major threats to human health [19,20]. A recent study showed that knockdown of both *per* and *tim* in *An. gambiae* and *An. stephensi* caused a decrease in swarming and mating rate [21]. In *Ae. aegypti*, knockout of *cyc* was found to reduced mating success [22]. However, it remains unclear how the molecular clocks regulate the mating behavior rhythm in *Ae. albopictus* and *Cx. quinquefasciatus*, two mosquito species with different activity patterns. Mosquito-control strategies, such as sterile insect technique (SIT) [23], that rely on manipulating or exploiting mating behaviors are rapidly becoming important tools in the management of mosquito populations. An in-depth understanding of the temporal characteristics of mating behavior and their mechanisms in these two species may provide new insights into mosquito control strategies.

Cuticular hydrocarbons (CHCs) are widely used as contact pheromones by insect during sexual communication [24,25]. CHCs are waxy molecules derived from fatty acids through a biosynthetic process involving desaturases, elongases, fatty acid synthases, and cytochrome P450 enzymes [26–28]. Studies in *Drosophila* and *Anopheles* show that the desaturase gene *desat1* is rhythmically expressed and regulates the production of cuticular hydrocarbons [21,29]. Little is known of the functions of CHCs and *desat1* genes in *Ae. albopictus* and *Cx. quinquefasciatus*.

We compare here the daily mating activity of *Ae. albopictus* and *Cx. quinquefasciatus* at discrete intervals under different light / dark conditions. Knockout strains of the core clock gene *clock* (*clk*) orthologs were established in both *Ae. albopictus* and *Cx. quinquefasciatus* using CRISPR/Cas9 to examine the relationship between the molecular clock and mating behavior rhythm of the two mosquito species. The daily mating activity between the mutant and wild-type (WT) strains were compared. Furthermore, the expression of the gene *desat1* orthologs was inhibited in *Ae. albopictus* and *Cx. quinquefasciatus* and their effects on CHCs synthesis and mating activity were analyzed.

## Materials and methods

### Mosquitoes strains

The *Ae. albopictus* (Foshan strain) and *Cx. quinquefasciatus* (Guangzhou strain) WT strains were obtained from the Center for Disease Prevention and Control (CDC), Guangdong Province, China. The *clk* mutants (*Aalclk*$^{\Delta 293}$ and *Cxqclk*$^{\Delta 98}$) were established and maintained by our laboratory. All mosquitoes were maintained under standard insectary conditions of 27 ±1˚C, 70±10% relative humidity, and a light/dark (LD) cycle of 12h/12h (250 lux light). To obtain experimental mosquitoes, the larvae (200 larvae/L water) were reared in stainless steel trays containing dechlorinated water and were provided daily with yeast and turtle food. Pupae of both species were collected individually and secured in 2mL Eppendorf tubes with 1 mL water. When adults emerged, virgin males and females were placed separately in paper bowls (9.5 × 6.7 × 6.2 cm$^3$) with a mesh cover and offered a 10% sucrose solution on a cotton wick.

### Mating activity analysis

Adult mosquitoes aged 2 days post emergence were placed in chambers with different light and dark conditions for four-days entrainment period. The 12:12 light/dark (LD) cycle refers

to lights on at zeitgeber time (ZT) 0 (8:00 am) and lights off at ZT12 (20:00 pm). The 12:12 dark/light (DL) cycle has lights on at ZT12 (20:00 pm) and lights off at ZT0 (8:00 am). The daily mating activity of mosquitoes was assessed on day 4 of LD and DL. In order to measure the mating activity during the free running period, the mosquitoes were measured on day 2 of constant darkness (DD) [12] after 4 days entrainment in LD (LDDD) and DL (DLDD). Mosquitoes were handled under dim red illumination during the nighttime conditions. At the beginning of each experiment, 10 virgin male and 10 virgin female mosquitoes were placed in a 250ml paper cup within an environment chamber and maintained at conditions of 27±1˚C, 70±10% relative humidity. Mosquitoes were allowed to mate for 3 hours under different light and dark conditions prior to dissecting spermathecae of females. The mating rate was calculated as the percentage of the number of inseminated females divided by the number of dissected females. A new group of 10 virgin males were transferred into another 250ml paper cup containing 10 virgin females to mate for 3h at the next zeitgeber time. Similar experiments were performed at different zeitgeber times of a day. Both *Ae. albopictus* and *Cx. quinquefasciatus* were performed mating experiments on the same procedure and the mating activity experiments were repeated three times. Under LD or LDDD condition, the total mating rate during daytime or subjective daytime (ZT0-12) was calculated as the number of inseminated females divided by the number of dissected females at zeitgeber times including ZT0-3, ZT3-6, ZT6-9 and ZT9-12. The total mating rate during nighttime (ZT12-24) was calculated as the number of inseminated females divided by the number of dissected females at zeitgeber times including ZT12-15, ZT15-18, ZT18-21 and ZT21-24. Under DL or DLDD conditions, the calculation of total mating rates for daytime and nighttime was the opposite of that for LD or LDDD.

### *Clock*-knockout generated using CRISPR/Cas9

Injections into phenotypically wild-type embryos were performed using the *Ae. albopictus* Foshan strain and *Cx. quinquefasciatus* Guangzhou strain. In order to produce a large deletion in the genome, a dual sgRNA knockout strategy was used. CRISPOR (http://crispor.tefor.net), an online tool, was uesd to identify appropriate guide RNAs (sgRNAs) target sequence containing the T7 polymerase-binding site and specifically targeting 20 bp of the site. The exon 2 of *Ae. albopictus* and exon 3 of *Cx. quinquefasciatus* were used to design and determine optimum candidate sgRNAs, respectively. According to the off-target analyses, top two, sgRNA1 and sgRNA2, were selected and these target *Aalclk* at positions 37 / 248 for *Ae. albopictus* and target *Cxqclk* at positions 170 / 323 for *Cx. quinquefasciatus*. The injection mixes contained 300 ng/μL TrueCut Cas9 Protein v2 (Thermo Fisher Scientific), 100 ng/μL purified sgRNA1 and 100 ng/μL purified sgRNA2 added to PBS (PH = 7.2). Embryo microinjections of the two mosquito species were used procedures described for mosquitoes [30].

### DNA extraction, *clock* mutation detection and screening

The larvae hatched from all $G_0$ generation embryos after microinjection were reared to adult mosquitoes. Male and female had been separated from each other in the pupal stage to avoid mating. We designed oligonucleotide primers complementary to genomic DNA sequences at the 5'-end (upstream) and 3'-end (downstream) of the sgRNA target sites for screening mutations. Genomic DNA was extracted from legs of individual $G_0$ adults using the Extract-N-Amp Tissue PCR Kit (Sigma-Aldrich) following the manufacturer's protocol. Mosaic $G_0$ mosquitoes were screened by gene amplification (polymerase chain reaction, PCR) by using Maxima Hot Start Green PCR Master Mix (Thermo Scientific). Subsequently, each mosaic $G_0$ virgin male was mated with three wild-type virgin females. Meanwhile, each mosaic $G_0$ virgin

female was mated with three wild-type virgin males. Female mosquitoes were fed blood to lay eggs and $G_1$ generation was obtained (S1 Table). The genomic DNA was extracted from legs of individual $G_1$ adult mosquitoes and the heterozygous $G_1$ adults were detected by PCR. The PCR products were purified with a MiniBEST DNA Fragment Purification Kit (Takara) and subcloned in the PMD-18 T Vector (Takara). Sequences of individual clones were detected by Sanger sequencing and sequence analysis were performed with MEGA7 (version 7.1.0, http://www.megasoftware.net/). Through sequence analysis, the $G_1$ adults heterozygous male and female mosquitoes with the same mutation type were screened and crossed with each other to produce $G_2$ generation. The homozygous *clk* mutants were selected from $G_2$ generation by the same detection and sequence analysis. These homozygous *clk* mutants were mass-crossed to establish the *clk* mutant strain (*Aalclk*$^{\Delta 293}$ and *Cxqclk*$^{\Delta 98}$). Before testing, genomic DNA was extracted from *Aalclk*$^{\Delta 293}$ and *Cxqclk*$^{\Delta 98}$ samples using a MiniBEST Universal Genomic DNA Extraction Kit (Takara). Mutant strains were confirmed to be homozygous with PCR (S1 Fig). All of the designed sgRNAs were checked using Cas-OFFinder online software (http://www.rgenome.net/cas-offinder/) to predict potential off-target sites (S2 Table). Off-target effects were tested in genomic DNA extracted from *clk* mutant strains. No off-target effect was confirmed by PCR and sequence analysis (S2 Fig). All primers are listed in S5 Table.

## RNA isolation and quantitative real-time PCR analysis

Total RNA from 10 adult mosquitoes heads or bodies at different zeitgeber times on day 4 under 12:12 LD cycles was isolated using the TRIzol Reagent (Life Technologies, Carlsbad CA, USA) according to the manufacturer's instructions. Each treatment was replicated three times. The RNA quantity and quality were determined using a NanaDrop 2000 Spectrophotometer (Thermo Scientific). A total of 5 μg RNA was digested using TURBO DNA-free Kit (Life Technologies, Carlsbad CA, USA) to remove genomic DNA following the manufacturer's instructions. First-strand cDNA was synthesized with GoScript Reverse Transcription System (Promega Corporation, Madison WI, USA) following the manufacturer's instructions. Gene expression was assessed by quantitative real-time PCR analysis with the SYBR selected master mix (Life Technologies). PCR involved an initial denaturation at 95˚C for 2 min, 40 cycles of 15 sec at 95˚C, 15 sec at 55˚C, and a final extension at 72˚C for 1min. All primers are listed in S5 Table.

## dsRNA-mediated gene silencing and mating assessment in adult mosquitoes

Double-stranded RNA (dsRNA) was synthesized using T7 RiboMAX Express RNAi System (Promega, Madison, WI, USA), as described elsewhere for mosquitoes [31]. The coding region fragments of *desat1* (LOC109412434, LOC6047469) were amplified from *Ae. albopictus* and *Cx. quinquefasciatus* cDNA with forward and reverse primers containing the T7 promoter sequence at their 5' ends (GGATCCTAATACGACTCACTATAGG) (S5 Table). The PCR products was purified with the GeneJET PCR Purification Kit (Thermo Fisher Scientific, Carlsbad CA, USA) and the purified PCR products were used as templates to synthesize dsRNA. The dsRNA for green fluorescent protein GFP was synthesized and used as negative control for the non-specific dsRNA effects. Two-day-old virgin *Ae. albopictus* and *Cx. quinquefasciatus* adults were anesthetized on a cold tray, and ~ 500nL of dsRNA solution (3000ng/μL) was injected laterally into the thorax under a microscope. Each treatment was replicated three times with 30 male or 30 female mosquitoes per replicate. The RNAi silencing efficiency was assessed over five consecutive days post injection (dpi). At dpi 4, which was the most efficient, 30 injected males or females were exposed to 30 virgin females or males in paper bowls

$(9.5 \times 6.7 \times 6.2 \text{ cm}^3)$. Mating assays were started at ZT9 and allowed to continue for 24 hours. Mating activity analysis was also assessed at each zeitgeber time using the same method as above. The ribosomal protein S7 genes of *Ae. albopictus* and *Cx. quinquefasciatus* were used as an endogenous control. The primers used for dsRNA synthesis and gene detection are shown in S5 Table.

## Extraction of cuticular hydrocarbons

Cuticular hydrocarbons (CHCs) were extracted from 80 virgin male mosquitoes injected with *GFP* and *desat1* dsRNA at dpi 4, respectively. Male mosquitoes were soaked in 800 μL hexane for 5 min in 10 mL conical glass centrifuge tubes with centrifugation at 2,500×g. Each groups were treated at the same time point ZT12 under 12:12 LD cycles. The extract was pipetted to a new 2mL conical glass tubes and concentrated under a gentle stream of nitrogen gas. The concentrated extracts were immediately dissolved in 200 μL hexane before their injection into the Gas Chromatography/Mass Spectrometer (GC/MS). Alkanes standards (from C7-C40) O2si was purchased from Anpel company. Six different concentrations including 0.1ppm, 0.5ppm, 1 ppm, 5ppm, 10ppm and 50ppm of n-alkane standards (C7-C40) were injected with every set of samples. Samples were analyzed within 24 h of preparation.

## GC/MS analysis of cuticular hydrocarbons

Identification and quantifications of CHCs were performed by using a GC/MS instrument (7890B-7000D, Agilent, USA), equipped with a HP-5MS column (Agilent, 30 m × 0.25 mm i. d., 0.25μm film thickness, 5% phenyl methyl siloxane stationary phase). The GC/MS conditions were as follows: the GC injection port temperature was maintained at 300˚C and the samples were injected in the split-less mode. The oven temperature was programmed from an initial temperature of 80˚C held for 2min, followed by a ramp at 20˚C/min to 200˚C, and a second ramp at 5˚C/min to 290˚C and held in that temperature for 19 min. The transfer line between GC and MS was set at 290˚C, and the temperatures of the MS source were 230˚C and operated under electron ionization mode. The scan mode was used for monitoring the targeted compounds. The carrier gas was helium (purity > 99.999%) at a constant flow rate of 1.0 mL/min. Data acquisition and evaluation were carried out using Agilent MassHunter Data Acquisition, Quantitative Analysis and Qualitative Analysis programs (version B07.00, Agilent Technologies, CA), respectively. The peaks were identified based upon comparison with standards and the National Institute of Standards and Technology library (NIST, version 17). The peak area of each CHC component was calculated as a proportion of the total area of all detected peaks for each samples.

## Effect of cuticular hydrocarbons on mating activity

Hydrocarbons were dissolved in n-hexane at a concentration of 75 μg/mL [21] and paintbrushes were used to apply 1μL on the abdomen of two-day-old virgin male *Ae. albopictus* and *Cx. quinquefasciatus* mosquitoes. Mosquitoes which received no solvent (blank group) and those that only applied the n-hexane (control group) were included for comparisons. Hydrocarbons were applied to *Ae. albopictus* and *Cx. quinquefasciatus* males at ZT6. Three hours after perfuming, 20 treated male and 20 virgin female *Ae. albopictus* adults were transferred into paper bowls and allowed to mate for 2 hours. The mating time was from ZT9 to ZT11, which was the peak of mating activity. For *Cx. quinquefasciatus*, 20 treated male and 20 virgin female adults were transferred into paper bowls at ZT9 and left overnight. Spermathecae were then dissected from females and examined for insemination status. The different mating durations for *Ae. albopictus* and *Cx. quinquefasciatus* were used to try to achieve the mating rate at

an intermediate level to help determine if CHCs promote or inhibit mosquito mating. The mating assays were repeated three times. Treated mosquitoes were maintained on 10% sucrose at 27±1˚C and 70±10% relative humidity, with a 12:12 LD cycle.

### Statistical analysis

All statistical analyses were performed using SPSS version 27.0 (IBM SPSS Statistics). All data were first tested to determine they followed a normal distribution by using a Shapiro-Wilk normality test ($\alpha = 0.05$). To compare the mating rates at different zeitgeber times, the generalized linear model (GLM) of binomial distribution with logic link and Sidak's test for pairwise comparisons were used. To compare the differences in mating rates between the two groups, GLM of binomial distribution with logic link was also used. The relative mRNA levels between the two groups at a certain time were compared using the Student t-test. The differences in the relative mRNA levels of *desat1* gene at different zeitgeber times were compared using a one-way ANOVA. The Student t-test was also used to compare differences in the relative percentages of each CHC components between treatments and controls. A value of $P < 0.05$ was considered to be statistically significant.

## Results

### *Aedes albopictus* and *Cx. quinquefasciatus* display distinct mating activity rhythms

The diel mating activities of *Ae. albopictus* and *Cx. quinquefasciatus* were measured during 12:12 LD cycles with lights on at ZT0, lights off at ZT12 (Fig 1A). *Ae. albopictus* mating occurs throughout the LD cycles, however, the mating activities were significantly influenced by the time of a day (GLM, $\chi^2 = 143.00$, d.f. = 7, $P < 0.0001$). Mating activities peaked at ZT0-3 and ZT9-12 in *Ae. albopictus* (Fig 1B). To determine whether this diel mating rhythms of *Ae. albopictus* were controlled by an endogenous clock, the mating activities were measured on day 2 of constant dark (DD) after 4 days of entrainment in LD cycles (LDDD). Mating activity rhythm was still observed under LDDD (GLM, $\chi^2 = 35.15$, d.f. = 7, $P < 0.0001$) and was similar to the LD cycles with mating activity peaking at CT0-3 and CT9-12 (Fig 1B). In addition, we reversed the onset of light and dark (lights on at ZT12 and lights off at ZT0, DL) and found that the mating activity rhythm appeared entrained by the DL cycles (Fig 1C). The mating rate of *Ae. albopictus* was high during the daytime and low at night, which was similar to LD cycles. The mating peaks occurred at ZT12-15 and ZT21-24 in DL. In order to demonstrate circadian entrainment of mating rhythm rather than acute stress response to light, the mating activities of mosquitoes on day 2 of DD after 4 days of entrainment in DL cycles (DLDD) were measured. The mating rhythm of mosquitoes under DLDD was similar to those of mosquitoes under DL cycles (Fig 1C). These results support the conclusion that the mating activity rhythm of *Ae. albopictus* is controlled by an endogenous clock and can be entrained to daily LD cycles.

Using the same procedures, the diel mating activities of *Cx. quinquefasciatus* were also significantly influenced by the time of a day (GLM, $\chi^2 = 21.86$, d.f. = 7, $P = 0.0027$) under LD cycles. However, *Cx. quinquefasciatus* rarely mated during the daytime and there was a single mating peak at ZT12-15 at night (Fig 1D). The mating activity rhythm remained consistent under LDDD as well as in LD with a mating peak at CT12-15 (Fig 1D). The mating rhythm of *Cx. quinquefasciatus* under DL cycles showed that most mosquitoes mated at night and there was only a mating peak at ZT0-3 (Fig 1E). The mating activity rhythm also remained consistent under DLDD as well as in DL (Fig 1E). The mating rhythm of *Cx. quinquefasciatus* is also controlled by an endogenous clock and can be entrained to daily LD cycles. These results

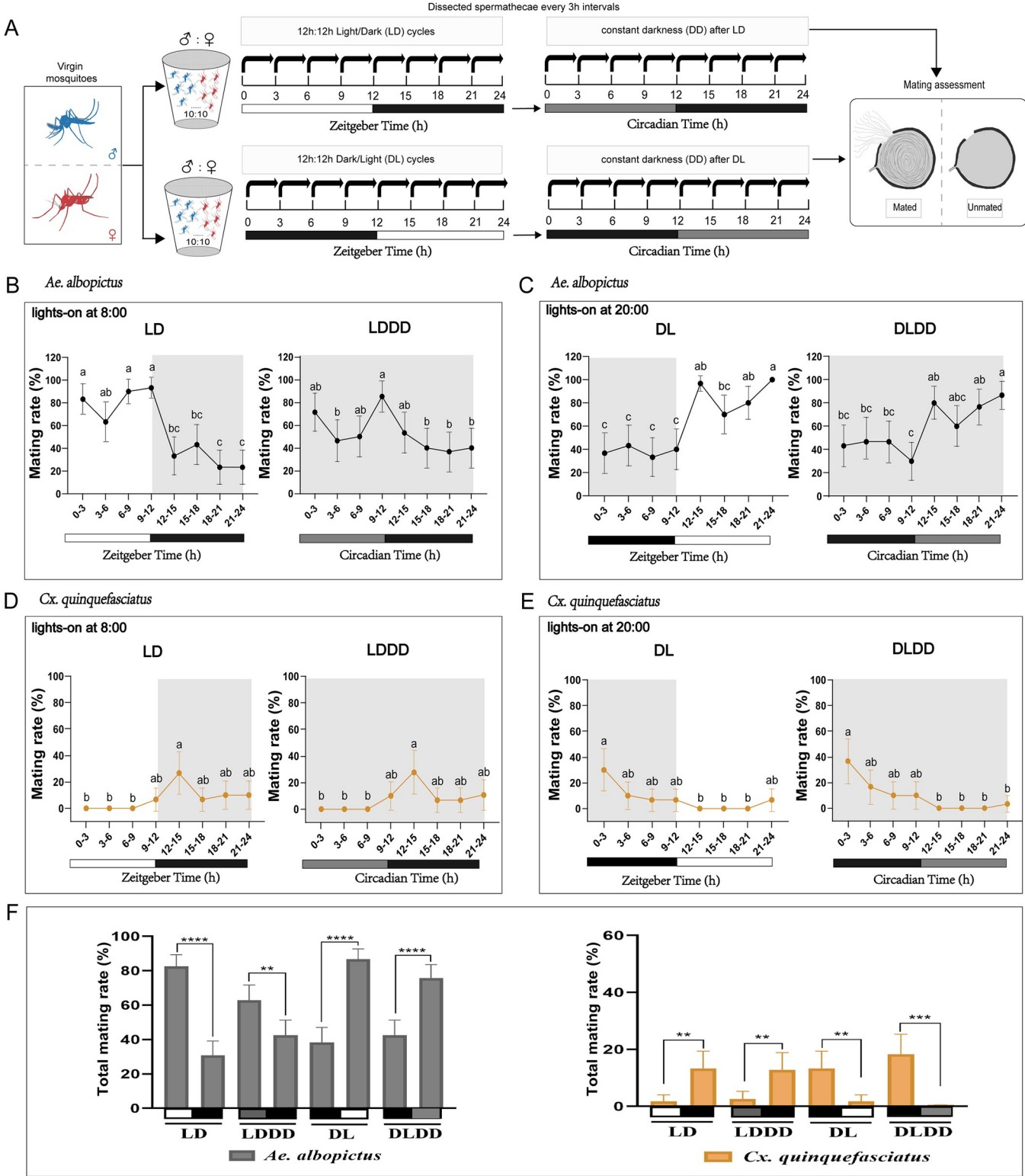

**Fig 1. Mating activities of *Ae. albopictus* and *Cx. quinquefasciatus* at different times of the day under different light conditions.** (**A**) Experimental flow chart of mating rate of *Ae. albopictus* and *Cx. quinquefasciatus* at different zeitgeber times in a day. Blue represents male mosquitoes and red represents female mosquitoes. (**B**) Daily changes in mating activities of *Ae. albopictus* on day 4 under 12:12 light/dark (LD) cycles (lights on at ZT0) and mating activities on day 2 under constant dark (DD) after 4 days of entrainment in LD cycles. (**C**) Daily changes in mating activities of *Ae. albopictus* on day 4 under 12:12 dark/light (DL) cycles (lights on at ZT12) and mating activities on day 2 under DD after 4 days of entrainment in DL cycles. (**D**) Daily changes in mating activities of *Cx.*

*quinquefasciatus* on day 4 under 12:12 LD cycles and mating activities on day 2 under DD after 4 days of entrainment in LD cycles. **(E)** Daily changes in mating activities of *Cx. quinquefasciatus* on day 4 under 12:12 DL cycles and mating activities on day 2 under DD after 4 days of entrainment in DL cycles. **(F)** Total mating rates of *Ae. albopictus* and *Cx. quinquefasciatus* during daytime and nighttime under different light conditions. The white portion of the bar below the graph indicates lights-on, the black portion indicates lights-off, and the gray portion indicates "subjective day" under DD condition. The black and orange lines represent *Ae. albopictus* and *Cx. quinquefasciatus*, respectively. Statistics were performed using generalized linear models (GLM) with binomial distribution and Sidak's test for pairwise comparisons. Values with different letters were significantly different between zeitgeber times. Error bars represent 95% confidence intervals (CIs). Each mosquito was measured only once. n = 28–30 for each zeitgeber time. $^{**}p < 0.01$, $^{***}p < 0.001$, $^{****}p < 0.0001$.

suggest that there is different mating activity rhythm of *Cx. quinquefasciatus* compared with *Ae. albopictus*. The two mosquito species have distinct and different peaks of mating activity.

The total mating rates during daytime or subjective daytime (ZT0-12) and nighttime (ZT12-24) were calculated to evaluate the difference in mating activity between day and night under LD or LDDD condition. For DL or DLDD, the calculation of total mating rates during daytime and nighttime was the opposite of that for LD or LDDD. These combined results showed that the total mating rates of *Ae. albopictus* were significantly higher in the daytime or subjective daytime than that at night under different conditions (GLM, all $P < 0.01$; Fig 1F). In *Cx. quinquefasciatus*, the total mating rates were significantly higher at night than that in the daytime or subjective daytime under different conditions (GLM, all $P < 0.01$; Fig 1F).

## Cas9/guide RNA-mediated *clock* (*clk*) gene knockouts affect mating activity rhythms of *Ae. albopictus* and *Cx. quinquefasciatus*

Dual small guide RNAs (sgRNA) (sgRNA37/sgRNA248 for *Ae. albopictus*; sgRNA170/ sgRNA323 for *Cx. quinquefasciatus*) and Cas9 endonuclease were used to generate *clk* ortholog knockout mutations in the respective species following microinjection into embryos. The sgRNA target sites are in exon 2 in *Ae. albopictus* (Fig 2A) and exon 3 in *Cx. quinquefasciatus* (Fig 2F). The *Ae. albopictus* homozygous strains, *Aalclk*$^{\Delta293}$, has a deletion 293 base-pairs (bp) in length and a 76bp insertion and the *Cx. quinquefasciatus Cxqclk*$^{\Delta98}$ strain has a 98bp deletion and 8bp insertion, both resulting in frameshift mutations of the respective target genes (Figs 2A and 2F and S1). *clk* gene transcription products were measured under LD cycles in *Aalclk*$^{\Delta293}$, *Cxqclk*$^{\Delta98}$ and wild-type (WT; *Aalclk*$^+$ and *Cxqclk*$^+$) using qRT-PCR. The results showed strong fluctuations in mRNA abundance levels in the both WT controls, while *clk* knockout strains exhibited significantly reduced mRNA levels throughout the day (Fig 2B, 2C, 2G and 2H). These results support the conclusion that *clk* gene knockouts suppress the expression levels of the respective transcripts.

*Aalclk*$^{\Delta293}$ and *Cxqclk*$^{\Delta98}$ mating activity rhythms after 4 days of entrainment in LD cycles was observed to determine if it is impacted by disruptions of the respective *clk* genes. The mating rate of *Aalclk*$^{\Delta293}$ males was > 70% at each zeitgeber time during the day. However, there was a mating peak at ZT12-15 in dark phase, showing a delayed mating peak phase that did not occur in the WT group (Fig 2D). The total mating rate of *Aalclk*$^{\Delta293}$ at night (45.8%) was significantly higher than that of WT group (30.8%; GLM, $\chi^2 = 5.65$, d.f. = 1, $P = 0.017$), but there was no significant difference between *Aalclk*$^{\Delta293}$ and WT group in daytime mating rates (GLM, $\chi^2 = 1.61$, d.f. = 1, $P = 0.205$) (Fig 2E).

*Cxqclk*$^{\Delta98}$ mating activities were observed at each zeitgeber time during the daytime but not at night and the nighttime mating peak at ZT12-15 disappeared (Fig 2I). This was in contrast to the WT group, indicating that the mutation caused a significant phase change in the mating behavior in *Cxqclk*$^{\Delta98}$. Comparisons of the total mating rate during LD conditions revealed that the total mating rate of *Cxqclk*$^{\Delta98}$ at night (0.0%) was significantly lower than that of WT group (13.3%; GLM, $\chi^2 = 18.46$, d.f. = 1, $P < 0.0001$). The total mating rate of *Cxqclk*$^{\Delta98}$ during

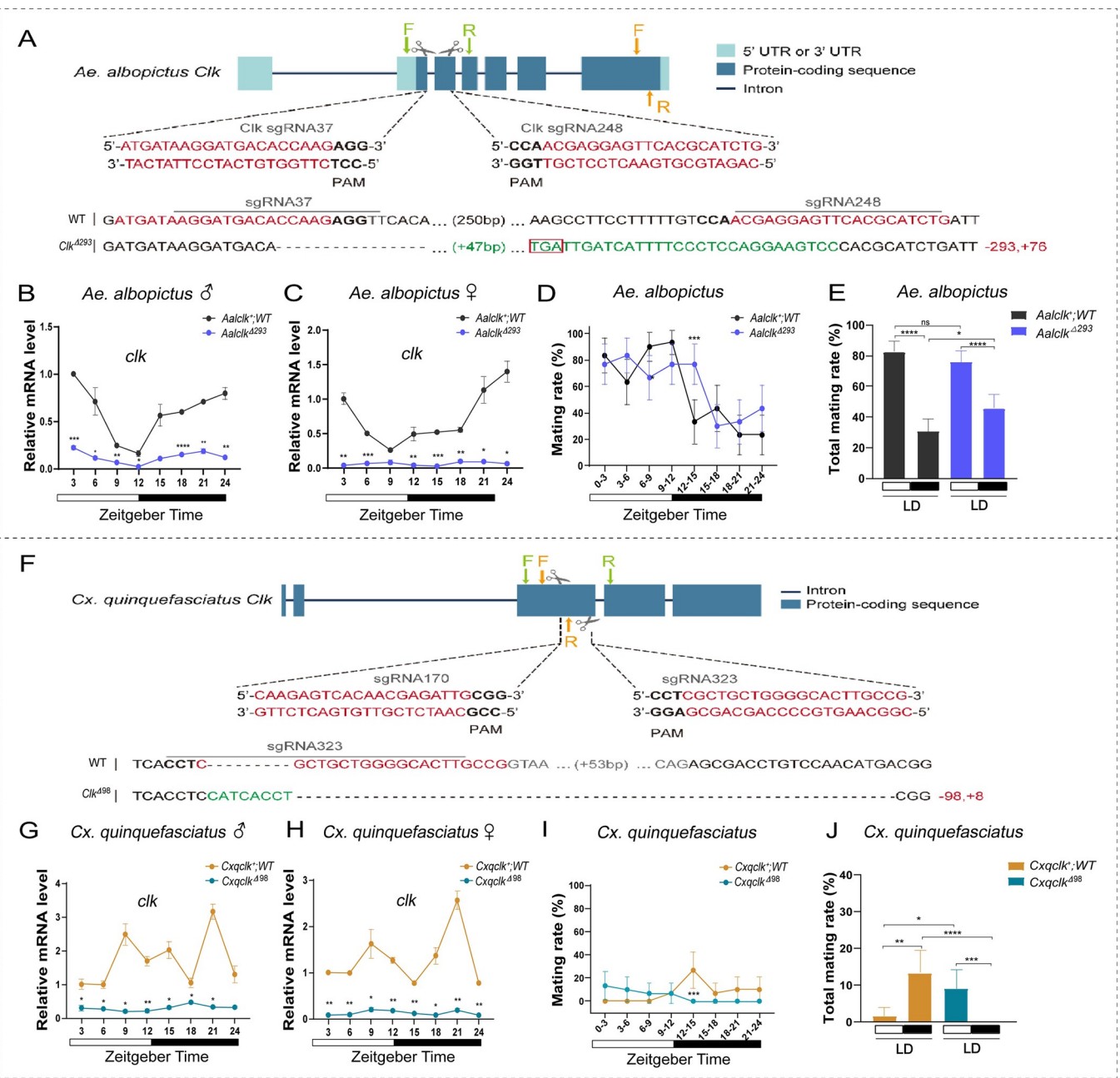

**Fig 2. CRISPR/Cas9-mediated *clk* mutagenesis and mating activities at different times of day in *clk* mutants.** (A) Schematic diagram of sgRNA-targeting sites to induce *clk* mutations in *Ae. albopictus* using CRISPR/Cas9 technology. The scissors represent the corresponding position of sgRNA in the genome. The red sequence is sgRNA and the gray sequence is intron sequence. The dotted line indicates base deletion and the green sequence indicates base insertion. The number in brackets indicates the number of omitted bases on the genome. The red box highlights the stop codon. The number at the end of the sequence indicates the number of inserted or missing bases on the genome. Green arrows indicate mutation identification primer positions, yellow arrows indicate the positions of qPCR detection primers. (**B-C**) Temporal expression patterns of *clk* mRNA between WT (*Aalclk*⁺) and *Aalclk*^Δ293^ under LD cycles. The abundance of *clk* mRNA was measured by qRT-PCR. Total RNA was extracted from 10 males or 10 females heads at 3h intervals. White and black bars indicate light phase and dark phase, respectively. (**B**) *Ae. albopictus* males, (**C**) *Ae. albopictus* females. (**D**) Daily changes in mating activities between WT (*Aalclk*⁺) and *Aalclk*^Δ293^ under LD cycles. (**E**) Total mating rates between WT (*Aalclk*⁺) and *Aalclk*^Δ293^ during daytime and nighttime. (**F**) Schematic diagram of sgRNA-targeting sites to induce *clk* mutations in *Cx. quinquefasciatus* using CRISPR/Cas9 technology. (**G-H**) Temporal expression patterns of *clk* mRNA between WT (*Cxqclk*⁺) and *Cxqclk*^Δ98^ under LD cycles. (**G**) *Cx. quinquefasciatus* males, (**H**) *Cx. quinquefasciatus* females. (**I**) Daily changes in mating activities between WT (*Cxqclk*⁺) and *Cxqclk*^Δ98^ under LD cycles. (**J**) Total mating rates between WT (*Cxqclk*⁺) and *Cxqclk*^Δ98^ during daytime and nighttime. For the relative mRNA level, statistics were performed using Student t-test and error bars represent the standard error of mean (SEM). Each treatment was replicated three times. For mating rate between two groups at a certain time, statistics were performed using GLM with binomial distribution and error bars represent 95% confidence intervals (CIs). Each mosquito was measured only once. n = 28–30 for each point. * $P < 0.05$, ** $P < 0.01$, ***$P < 0.001$, **** $P < 0.0001$.

daytime (9.2%) was significantly higher than that of WT group (1.7%; $\chi^2$ = 5.23, d.f. = 1, $P$ = 0.022) (Fig 2J).

*clk* mutation led to a phase disruption in the diel mating rhythm under LD cycle, resulting in *Ae. albopictus* showing a nighttime mating peak, and *Cx. quinquefasciatus* showing mating during the daytime but not at night. These results indicate that the core clock gene *clock* (*clk*) in the two mosquito species have significant roles in maintaining the diel mating activity rhythms.

## The temporal expression of *desat1* gene is different and regulated by *clk* in *Ae. albopictus* and *Cx. quinquefasciatus*

A recent study showed that the *desat1* gene was involved in the synthesis of pheromone and affected mating behavior in *An. gambiae* and *An. stephensi* [21]. As discussed previously, *Ae. albopictus* and *Cx. quinquefasciatus* had different mating activity rhythm. qRT-PCR analyses were used to characterize *desat1* transcript abundance oscillations after 4 days of entrainment in LD cycles. The results showed that *desat1* transcript levels in the heads and bodies of *Ae. albopictus* exhibited oscillations with expression peaks in both day and night (S3 Fig). The *Cx. quinquefasciatus desat1* transcripts had a different oscillation trend with peak abundance at the moment of the light/dark transition (S3 Fig). These results indicated that the relative abundance of *desat1* in *Ae. albopictus* and *Cx. quinquefasciatus* had contrasting and opposite trends of daily oscillation.

Mutations of *clk* gene orthologs had effects on the temporal abundance profiles of the respective *desat1* transcripts in two mosquito species. The abundance of *desat1* transcripts in *Aalclk*$^{\Delta293}$ males was high during light/dark transition at ZT12. The peak and trough values were out of phase compared to WT controls (Fig 3A). The *desat1* transcripts profile in *Cxqclk*$^{\Delta98}$ males was also out of phase and had lowest abundance during light/dark transition which was opposite to that of WT controls (Fig 3C). In contrast, the *desat1* transcript abundance profile of *Aalclk*$^{\Delta293}$ females (Fig 3B) and *Cxqclk*$^{\Delta98}$ females (Fig 3D) only changed in amplitude and there was no distinct phase change in temporal expression trend compared to WT controls. These results support the conclusion that the *clk* gene orthologs are involved in regulating the oscillatory expression of their respective *desat1* genes.

## *desat1* gene orthologs affect mating behavior and regulate cuticular hydrocarbons(CHCs) biosynthesis in *Ae. albopictus* and *Cx. quinquefasciatus*

Double-strand RNA (ds RNA) injections were used to modulate *desat1* transcript levels in virgin *Ae. albopictus* and *Cx. quinquefasciatus* males and qRT-PCR was used to determine the gene silencing efficiency over five consecutive days post injection (dpi). The lowest transcript abundance was detected at 4 dpi in both species (Fig 3E and 3F). The effects of *desat1* on mosquito mating activity and mating rhythm were then measured. Male mosquitoes on 4 dpi were mated with virgin females for 24 hours and mating rhythms of one day under 12:12 LD cycles were conducted using the previously described spermathecae dissection assay (Fig 3G). For the mating assay, 30 injected males on 4 dpi were exposed to 30 virgin females at ZT9 and they were exposed for 24h. The average mating rates of the *desat1* dsRNA injected males of both species (65.6%, *Ae. albopictus*; 62.2%, *Cx. quinquefasciatus*) were significantly reduced compared with *GFP* dsRNA controls (96.7%, GLM, $\chi^2$ = 18.83, d.f. = 1, $P$ < 0.0001, *Ae. albopictus*; 90.9%, GLM, $\chi^2$ = 17.61, d.f. = 1, $P$ < 0.0001, *Cx. quinquefasciatus*; Fig 3H and 3I). Virgin female mosquitoes also were injected with *desat1* dsRNA and it did not result in any differences in mating rates in either species when compared with *GFP* dsRNA-injected controls

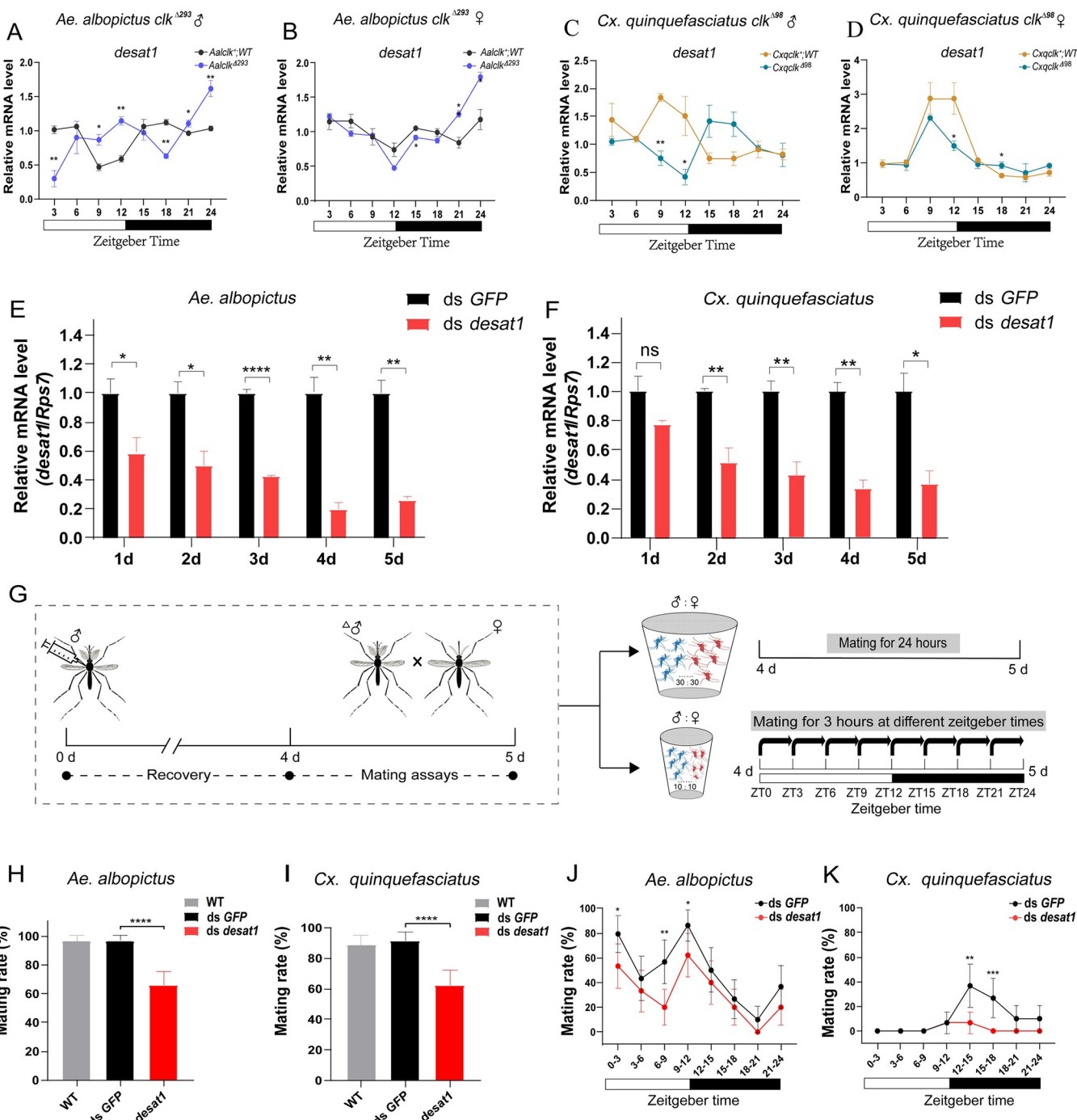

**Fig 3. Changes in temporal expression of *desat1* in *Ae. albopictus* and *Cx. quinquefasciatus* after *clk*-knockout and effects of *desat1* on mosquito mating activity.** (**A-D**) The expression pattern of *desat1* at different times of a day in WT and *clk* mutant mosquitoes heads under LD cycles. The housekeeping gene *RpS7* was used as the internal control for qRT-PCR. Data were normalized to median fold change. (**A**) WT (*Aalclk*[+]) and *Aalclk*[Δ293] males, (**B**) WT (*Aalclk*[+]) and *Aalclk*[Δ293] females, (**C**) WT (*Cxqclk*[+]) and *Cxqclk*[Δ98] males, (**D**) WT (*Cxqclk*[+]) and *Cxqclk*[Δ98] females. (**E-F**) Silencing efficiency of *desat1* of male *Ae. albopictus* (**E**) and *Cx. quinquefasciatus* (**F**) injected with *GFP* and *desat1* dsRNA. The X-axis represents days post thoracic injection (dpi) and Y-axis represents genes relative transcript accumulation. Statistics were performed using Student t-test and error bars represent the standard error of mean (SEM). Each treatment was replicated three times. (**G**) Experimental flow chart of the effect of *desat1* gene on mating behavior of *Ae. albopictus* and *Cx. quinquefasciatus*, and 'Δ' represents male mosquitoes injected with dsRNA. (**H-I**) Effects of silencing *desat1* expression on mating activity in male *Ae. albopictus* (**H**) and male *Cx. quinquefasciatus* (**I**). A total of 30 injected males on 4 dpi were exposed to 30 virgin females in paper bowls. Mating assay was started at ZT9 and they were exposed for 24h. Gray, black and red columns represent WT group, the injection group of *GFP* dsRNA and the injection group of *desat1* dsRNA, respectively. n = 88–90 for each treatment. (**J-K**) Effects of silencing *desat1* on mating rhythm of *Ae. albopictus* (**J**) and *Cx. quinquefasciatus* (**K**) under LD cycles. The black

lines represent the injection of *GFP* dsRNA and the red lines represent the injection of *desat1* dsRNA. For mating rate, statistics were performed using GLM with binomial distribution and error bars represent 95% confidence intervals (CIs). n = 29–30 for each point. Each mosquito was measured only once. * *P* < 0.05, ** *P* < 0.01, *** *P* < 0.001, **** *P* < 0.0001, ns = not significant.

(GLM, all *P* > 0.05; S4 Fig). For the mating rhythms assay, the daily mating activity of *desat1* dsRNA-injected *Ae. albopictus* males showed that the overall mating rhythm did not change with mating peaks at ZT0-3 and ZT9-12. However, the mating rate decreased, especially for ZT0-3, ZT6-9 and ZT9-12 (GLM, all *P* < 0.05; Fig 3J). In *Cx. quinquefasciatus*, the daily mating activity of *desat1* dsRNA-injected males was significantly reduced at ZT12-15 and ZT15-18 (GLM, all *P* < 0.01; Fig 3K), but the overall mating rhythm did not change. These results support the conclusion that *desat1* knockdown has no effect on the overall mating rhythm of *Ae. albopictus* and *Cx. quinquefasciatus*, but significantly reduces mating rate under LD cycles.

GC/MS was used to examine the role of *desat1* in regulating the biosynthesis of cuticular hydrocarbons (CHCs). Analyses of cuticular extracts of *Ae. albopictus* males on 4 dpi identified 25 major CHCs in both *GFP* dsRNA-injected controls and *desat1* dsRNA-injected treatments, respectively (Figs 4A and S5 and S3 Table). Analyses of cuticular extracts of *Cx. quinquefasciatus* males on 4 dpi identified 17 major CHCs in both controls and treatments, respectively (Figs 4B and S5 and S4 Table). DsRNA-mediated silencing of *desat1* resulted in significant reduction of nonadecane (C19), tricosane (C23), nonacosane (C29) and hentriacontane (C31) (Student t-test, all *P* < 0.05), while heptacosane (C27) significantly increased in *Ae. albopictus* (Student t-test, t = 4.30, d.f. = 4, *P* = 0.0126) (Fig 4C). Silencing *desat1* resulted in significant reduction of heneicosane (C21) and C23 in *Cx. quinquefasciatus* (Student t-test, all *P* < 0.05), while C27 and C29 were significantly increased (Student t-test, all *P* < 0.05) (Fig 4D). These results support the conclusion that *desat1* is involved in the production of CHCs in these two mosquito species.

Two day-old virgin male *Ae. albopictus* and *Cx. quinquefasciatus* were treated with hydrocarbons and applied to their abdomens at ZT6. Three hours after perfuming, 20 treated male and 20 virgin female *Ae. albopictus* mosquitoes were allowed to mate for 2 hours. For *Cx. quinquefasciatus*, 20 treated male mosquitoes and 20 virgin female mosquitoes were exposed overnight (Fig 4E). Applications of C19 and C23 to *Ae. albopictus* males resulted in a significant increase in the mating rate (GLM, all *P* < 0.05), while C27 lead to a significant decrease in average mating rate (28.3%) compared with control group (47.5%; GLM, $\chi^2$ = 4.55, d.f. = 1, *P* = 0.033). C29 and C31 did not alter mating activity of *Ae. albopictus* (GLM, all *P* > 0.05; Fig 4F). In *Cx. quinquefasciatus* males, only C23 significantly increased the average mating rate (79.7%) compared with control group (61.0%; GLM, $\chi^2$ = 4.78, d.f. = 1, *P* = 0.029), while the other altered pheromones did not affect mating activity of *Cx. quinquefasciatus* (GLM, all *P* > 0.05; Fig 4G).

## Discussions

An in-depth understanding of mosquito mating biology is essential for mosquito control strategies that rely on interfering with vector reproduction [23]. The results of these experiments described here demonstrate that *Ae. albopictus* and *Cx. quinquefasciatus* have distinct and different mating rhythms. Their diel mating rhythms are regulated by their respective orthologs of the core clock gene *clk* and the pheromone synthesis-related *desat1* gene under LD cycles. The differences in the temporal expression of *desat1* gene between the two mosquitoes may be one of the mechanisms leading to the different mating rhythms. These findings, which are consistent with studies in *An. stephensi* [21], support the conclusion that the core clock genes

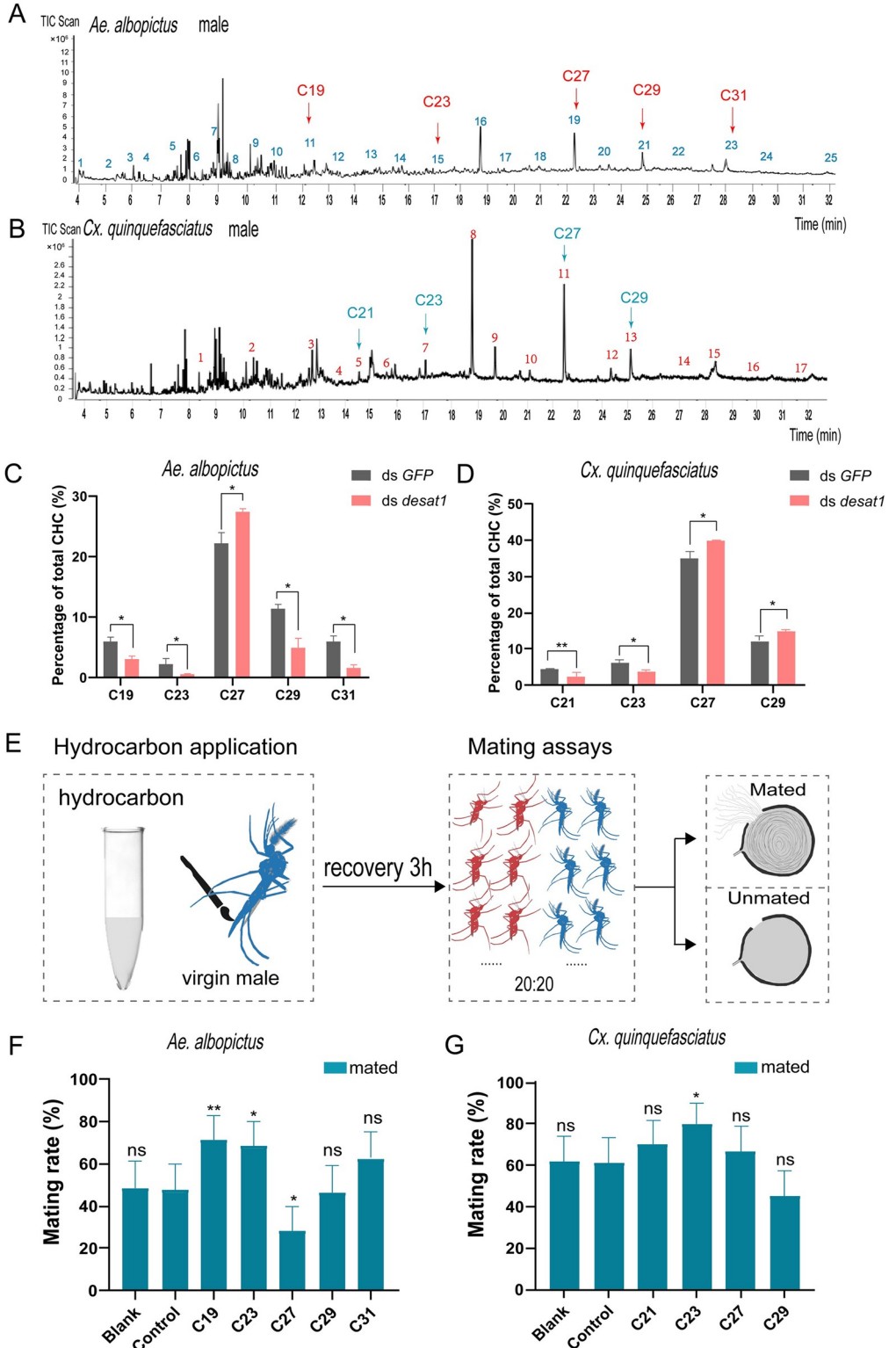

**Fig 4. Silencing of *desat1* changed the CHC profiles of male *Ae. albopictus* and *Cx. quinquefasciatus* and altered mating activity. (A-B)** Representative CHC profiles of male *Ae. albopictus* (**A**) and *Cx. quinquefasciatus* (**B**) on day 4 after *GFP* ds RNA injection. Numbers above the peaks correspond to peak numbers given in S3 and S4 tables. Comparison of CHCs between mosquitoes injected with ds *GFP* and ds *desat1* in *Ae. albopictus* (**C**) and *Cx. quinquefasciatus* (**D**). The gray bars represent the groups injected with *GFP* dsRNA, and the pink bars represent the

groups injected with *desat1* dsRNA. Statistics were performed using Student t-test and error bars represent the standard error of mean (SEM). Each treatment was replicated three times. **(E)** Flow chart of mating experiment after hydrocarbons application. Blue represents male mosquitoes and red represents female mosquitoes. Hydrocarbons were applied to *Ae. albopictus* and *Cx. quinquefasciatus* males at ZT6. Three hours after perfuming, 20 treated male mosquitoes and 20 virgin female mosquitoes were allowed to mate for 2 hours in *Ae. albopictus* or allowed to mate overnight in *Cx. quinquefasciatus*. **(F-G)** Effect of the CHCs on mosquito mating activity of *Ae. albopictus* (**F**) and *Cx. quinquefasciatus* (**G**). For mating rate, statistics were performed using GLM with binomial distribution and error bars represent 95% confidence intervals (CIs). Each mosquito was measured only once. n = 59–60 for each group, * *P* < 0.05, ** *P* < 0.01, ns = not significant.

regulate diel mating rhythm by controlling the temporal expression of pheromone-related genes. This mechanism may be conserved throughout mosquito species.

Circadian patterns, swarm sizes and mating activities vary widely among different mosquito species [32–34]. For example, *Ae. aegypti* mate in single pairs or in small swarms around the human host during the day [35,36], whereas *An. freeborni* swarm by the thousands and mates at dusk [34]. In this study, we found the mating behaviors of *Ae. albopictus* and *Cx. quinquefasciatus* showed distinct diel rhythm patterns under different light/dark cycle conditions. In LD cycles, *Ae. albopictus* displayed mating activity throughout the day but mating peaks were observed at ZT0-3 and ZT9-12, while *Cx. quinquefasciatus* rarely mated during the day and had a mating peak at ZT12-15. Experiments in constant dark and reversed light/dark cycles showed that endogenous mating rhythms existed in both species of mosquitoes and the mating rhythms could be reset by external light conditions. Notably, the mating rhythm may be different from the general motor activity rhythm. It has been reported that in cockroach *Leucophaea maderae* [37], the optic lobe receiving light signals contains oscillating factors that drive circadian motor activity, and ablation of part of optic lobe can lead to loss of detectable motor rhythm. However, cutting off the optic lobe partially in female cockroaches had no effect on mating rhythm, while similar ablations in males resulted in significant differences in mating time, but significant rhythm still existed. Although both motor activity and mating behavior are regulated by light synchronization and circadian clock, their output mechanisms may be different.

The circadian clock is controlled genetically, and mutation in'clock genes'can change rhythmic behaviour in animals, including insects, humans and other species. The molecular clock is thought to control the expression of output genes throughout the body, thereby temporally controlling the behavior [5]. In this study, knocking out the *clock* (*clk*) gene orthologs was found to have different effects on the mating behavior of *Ae. albopictus* and *Cx. quinquefasciatus*. *Aalclk* mutation affected the mating rhythm of diural *Ae. albopictus* under LD conditons. The mating peak delayed and appeared in the dark phase. However, there was still a mating activity rhythm in general. Whether the mating rhythm is directly affected by light exposure, or whether other core clock genes have compensatory effects needs to be further explored. In contrast, *Cxqclk* mutation significantly disrupted the mating rhythm of nocturnal *Cx. quinquefasciatus* under the same LD condition, suggesting that *clk* gene played an important role in the maintenance of the mating rhythm in this mosquito. Further research is needed to determine the effects of *clk* gene on other behavioral rhythms in both mosquito species. For mosquito disease vectors, clock genes expression and locomotor activity rhythms are well-documented [38–40]. Furthermore, some clock genes, such as *per*, *tim* and *cyc* have been found to be involved in locomotor activity, mating or blood feeding behaviors [21,22,41,42]. However, the complete molecular clock mechanisms of mosquitoes and how they regulate various behaviors requires further study. This will provide a significant scientific basis for vector biology and mosquito control.

The molecular pathways of mating rhythms may be more complex than those of motor rhythms [43], as the former involve temporal coordination of both males and females [14]. Sex pheromones related to mating behavior are proposed to be involved in the regulation of mating rhythms in cockroach [10]. In this study, we found significant differences in the temporal expression pattern of the pheromone synthesis-related *desat1* gene between *Ae. albopictus* and *Cx. quinquefasciatus*, and knockout of *clk* gene orthologs resulted in altered temporal expression of the *desat1* gene. The peak expression of *desat1* in *Aalclk*$^{\Delta 293}$ showed a phase delay while the expression of *desat1* in *Cxqclk*$^{\Delta 98}$ showed a phase inversion. These results support the conclusion that *clk* can regulate the mating rhythm of both species by controlling the temporal expression of *desat1* gene orthologs. The phase difference of the temporal expression of *desat1* may be one of the mechanisms for the distinct and different mating rhythms between *Ae. albopictus* and *Cx. quinquefasciatus*.

Knockdown of male *desat1* expression in both species significantly reduced mating rates, while knockdown of female *desat1* expression had no effect on mating. In mosquito mating system, mating is initiated by the male mosquitoes, and this is followed by mate recognition by the female mosquitoes during contact [44]. Recent study has shown that the desaturase gene *desat1* is up-regulated of swarming males and regulates the production of cuticular hydrocarbons [21]. Based on the role of male in mosquito mating system and the results in our study, we propose that *desat1* plays an important role in the CHCs synthesis in male mosquitoes and mating activity. However, whether there are other pheromone synthesis-related genes or receptors in female mosquitoes needs further research. Knockdown of *desat1* affected the mating rate of each zeitgeber time under the LD cycle, but the phase of the mating rhythm was not changed at all. It indicates that *desat1* gene does not affect the rhythm, but could act as a downstream gene to affect the mating behavior. The further mechanism of the interaction between *desat1*, clock genes and zeitgebers remains to be explored. In addition, we found that *desat1* expression peak and mating peak did not coincide. The former occurred earlier than the latter. This process may involve pheromone synthesis, mating activity initiation, mate recognition and other events.

Many insects use CHCs as species and sex recognition signals for mating [28,29,45]. In *Drosophila*, knockdown of *desat1* was sufficient or largely inhibited pheromone biosynthesis, and this effect was particularly pronounced in males [29,46,47]. In recent study of *Anopheles*, the *desat1* ortholog regulates the production of CHCs [21]. In this study, *desat1* knockdown in male *Ae. albopictus* resulted in decreases of nonadecane and tricosane which were found to promote mating, while resulted in an increase of heptacosane which was found to inhibit mating. In *Cx. quinquefasciatus*, *desat1* knockdown resulted in a significant decrease of tricosane, which was found to promote mating. Tricosane has been reported to be associated with mate recognition in *D. suzukii* [48] and promote mating in the male white-marked tussock moth, *Orgyia leucostigma* [49], as well as attracting ovipositing female houseflies, *Musca domestica* [50]. Nonadecane has an important role as the sex pheromone and allomone on several lepidopteran species [51–54]. Heptacosane is thought to facilitate the mating activity in *An. stephensi* [21] and the tea weevil, *Myllocerinus aurolineatus* [55], however, it also has been reported to have an inhibitory effect on reproduction in the wasp, *Vespula vulgaris* [56]. Previous studies showed that CHC abundance in male *D. melanogaster* varied with light and time [57,58], supporting the conclusion that male CHCs are a dynamic feature. These results support the conclusion that *desat1* gene orthologs play an important role in the biosynthesis of sex pheromones such as cuticular hydrocarbons in both *Ae. albopictus* and *Cx. quinquefasciatus*. Knockdown of *desat1* reduced or increased levels of certain cuticular hydrocarbons, which could promote or inhibit mating. The temporal expression of the *desat1* gene is regulated by the *clock* gene and this may be one of the mechanisms of mating rhythm formation.

## Conclusions

We can conclude from these findings that *Ae. albopictus* and *Cx. quinquefasciatus* have significantly different diel mating rhythms that involve the regulation of *clock-desat1*-CHCs pathway under LD cycles (S6 Fig). The *desat1* gene plays a role in the biosynthesis of cuticular hydrocarbons in both species, thus affecting mating. The temporal expression of *desat1* is regulated by *clk* gene orthologs. Clock genes make up the circadian clocks that entrain to external light signals to give temporal characteristics to mating behavior. The observed distinct and different mating activity rhythms of two mosquito species and their molecular mechanisms regulated by clock genes and sex pheromones, may provide the basis for developing precise mosquito control strategies.

## Supporting information

**S1 Fig. Identification of mutation in *Ae. albopictus* and *Cx. quinquefasciatus clk* mutant strains.** Genomic DNA extracted from *Ae. albopictus clk*$^{\Delta 293}$ and *Cx. quinquefasciatus clk*$^{\Delta 98}$ and mutations were confirmed with PCR.
(TIF)

**S2 Fig. Sequence analysis of the off-target site in *Ae. albopictus* and *Cx.quinquefasciatus clk* mutant strains.** Potential off-target mutations were screened in genomic DNA extracted from *Ae. albopictus clk*$^{\Delta 293}$ and *Cx. quinquefasciatus clk*$^{\Delta 98}$ mutants. Blue arrows indicate the off-target test primers. No off-target mutations were confirmed.
(TIF)

**S3 Fig. Temporal expression patterns of *desat1* gene in the heads and bodies of *Ae. albopictus* and *Cx. quinquefasciatus*.** (A) *Ae. albopictus* male heads, (B) *Ae. albopictus* male bodies, (C) *Ae. albopictus* female heads, (D) *Ae. albopictus* female bodies, (E) *Cx. quinquefasciatus* male heads, (F) *Cx. quinquefasciatus* male bodies, (G) *Cx. quinquefasciatus* female heads, (H) *Cx. quinquefasciatus* female bodies. Tissues were collected from 3 replicate groups with 10 individuals at each zeitgeber time with 3h intervals for 24 h under LD condition. RNA levels were quantified by qPCR. Each value was the mean±SEM. White and black shades represent the photophase and scotophase, respectively. *P*-value determined by one-way ANOVA.
(TIF)

**S4 Fig. Silencing *desat1* expression in *Ae. albopictus* and *Cx. quinquefasciatus* female mosquitoes does not affect mating acitivity.** (A) Silencing efficiency of *desat1* of female *Ae. albopictus* on 4dpi. (B) Mating rate of *Ae. albopictus*. (C) Silencing efficiency of *desat1* of female *Cx. quinquefasciatus* on 4dpi. (D) Mating rate of *Cx. quinquefasciatus*. A total of 30 injected females on 4 dpi were exposed to 30 virgin males at ZT9 and they were exposed for 24h. Statistics were performed using GLM with binomial distribution and error bars represent 95% confidence intervals (CIs). Each mosquito was measured only once. n = 88–90 for each group, $^{**}$ $P < 0.01$, ns = not significant.
(TIF)

**S5 Fig. (A-B) Representative CHC profiles of male *Ae. albopictus* (A) and *Cx. quinquefasciatus* (B) on day 4 after *desat1* ds RNA injection.** Numbers above the peaks correspond to peak numbers given in S3 Table and S4 Table.
(TIF)

**S6 Fig. A schematic model of *Ae. albopictus* and *Cx. quinquefasciatus* mating rhythms and their regulation by *clock-desat1*-CHCs pathway.**
(TIF)

**S1 Table. Statistics of mutation rates of $G_0$ adults and $G_1$ mutant pools in *Ae. albopictus* and *Cx. quinquefasciatus*.**
(DOCX)

**S2 Table. *clk* gRNA off-target sites in *Ae. albopictus* and *Cx. quinquefasciatus* genome.** Mismatches bases are highlighted in lowercase and red.
(DOCX)

**S3 Table. Identity of cuticular hydrocarbon peaks of male adult *Ae. albopictus*.**
(DOCX)

**S4 Table. Identity of cuticular hydrocarbon peaks of male adult *Cx. quinquefasciatus*.**
(DOCX)

**S5 Table. Primers used in this study.**
(DOCX)

## Acknowledgments

The authors would like to thank the Guangdong Provincial Center for Disease Control and Prevention for providing the *Ae. albopictus* Foshan strain and the *Cx. quinquefasciatus* Guangzhou strain for this study, Dr. Tong Liu for his technical support and guidance in CRISPR/Cas9, and other colleagues from Southern Medical University for their advice and assistance in this study.

## Author Contributions

**Conceptualization:** Shuang Liu, Jiayong Zhou, Xiao-Guang Chen.

**Data curation:** Shuang Liu, Jiayong Zhou, Ling Kong, Yiquan Cai, Hongkai Liu.

**Formal analysis:** Shuang Liu, Jiayong Zhou, Ling Kong.

**Funding acquisition:** Xiao-Guang Chen.

**Investigation:** Xiao-Guang Chen.

**Methodology:** Shuang Liu, Jiayong Zhou, Ling Kong, Yiquan Cai, Hongkai Liu, Zhensheng Xie, Xiaolin Xiao.

**Project administration:** Xiao-Guang Chen.

**Resources:** Xiao-Guang Chen.

**Software:** Shuang Liu, Jiayong Zhou.

**Supervision:** Anthony A. James, Xiao-Guang Chen.

**Validation:** Shuang Liu, Jiayong Zhou.

**Visualization:** Shuang Liu, Jiayong Zhou, Ling Kong.

**Writing – original draft:** Shuang Liu, Jiayong Zhou, Xiao-Guang Chen.

**Writing – review & editing:** Anthony A. James, Xiao-Guang Chen.

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
