## [Decision Letter · Decision Letter 0]

26 Aug 2022

Dear Professor Chen,

Thank you very much for submitting your manuscript "Clock genes regulate circadian mating activity in the vector mosquitoes, Aedes albopictus and Culex quinquefasciatus" for consideration at PLOS Neglected Tropical Diseases. As with all papers reviewed by the journal, your manuscript was reviewed by members of the editorial board and by several independent reviewers. In light of the reviews (below this email), we would like to invite the resubmission of a significantly-revised version that takes into account the reviewers' comments. 

The reviewers have completed their assessment of the paper and find that it is sufficient quality following revision for PLoS NTD. Please address the comments provided by the reviewers individually.

We cannot make any decision about publication until we have seen the revised manuscript and your response to the reviewers' comments. Your revised manuscript is also likely to be sent to reviewers for further evaluation.

Sincerely,

Joshua B. Benoit

Academic Editor

Alvaro Acosta-Serrano

Section Editor

The reviewers have completed their assessment of the paper and find that it is sufficient quality following revision for PLoS NTD. Please address the comments provided by the reviewers individually.

Reviewer's Responses to Questions

**Key Review Criteria Required for Acceptance?**

**Methods**

-Are the objectives of the study clearly articulated with a clear testable hypothesis stated?

-Is the study design appropriate to address the stated objectives?

-Is the population clearly described and appropriate for the hypothesis being tested?

-Is the sample size sufficient to ensure adequate power to address the hypothesis being tested?

-Were correct statistical analysis used to support conclusions?

-Are there concerns about ethical or regulatory requirements being met?

Reviewer #1: I have concerns regarding the use of statistical tests that are not optimized for handling frequencies and percentages. Please find suggestions for alternative tests in my comments to the authors.

Reviewer #2: Methods needs more details. Were the nighttime condition mosquitoes handled? Under dim red light? n # for experiments should be provided throughout.

For mating assays, how many groups were tested per experimental rep? Was power analysis performed?

How were the Clock mutants screened and confirmed? How were the target sites selected and were the mutant lines tested for off-target effects? Please provide details on how mutagenesis was determined and if the mosquitoes outcrossed before being tested.

Figure 3H&I and S2 B&D: More details of experimental conditions should be provided, including time (ZT) and duration of these experiments. How many females were examined?

What time of day was the CHC-applied mating assay performed for Ae. albopictus?

The authors nicely demonstrate here that there is a mating rhythm for both species tested. Authors should provide time of experiment for any mating assays performed.

Variance in data is not very clear throughout this manuscript.

Reviewer #3: (No Response)

**Results**

-Does the analysis presented match the analysis plan?

-Are the results clearly and completely presented?

-Are the figures (Tables, Images) of sufficient quality for clarity?

Reviewer #1: Yes.

Reviewer #2: Lines 93: Please elaborate on what the authors mean by ‘evolutionary mechanisms’ here.

When referring to time, Zeitgeber Time (ZT), rather than real time, should be provided.

Line 239: 3 hours before darkness is ZT9, not ZT12. In Figure 1B, mating seems to peak at light:dark transition at ZT12. Either way, this sentence needs clarification.

Line 240: “~3h after lights off”. Is there reason for “~” here? This should be referred to as ZT15. 

Line 243: Needs clarification. While the Ae. albopictus evening mating peak looks delayed by 3 hrs, the morning mating peak appears have advanced by 3 hrs and is less pronounced under DD condition compared to LD. Similarly, Culex seems to have a small morning mating peak at ZT0/24 under LD that is not observed under DD.

Lines 247-250: Is 4 days considered enough for re-entrainment to a complete anti-phasic condition (reversed schedule) before measuring mating frequency? The shift in timing of mating peak suggests that the animals may still be re-entraining.

Line 248: Needs clarification. What do the authors mean by at each time period? 

Line 250-252: Lack of mating after lights on in Culex maybe due to acute suppression rather than re-entrainment. 

In order to demonstrate circadian re-entrainment of mating rhythm, the assay must be followed by sufficient re-entrainment period and then performed under free running condition (DD).

Line 265: While I agree with this conclusion, the way that DD data in Fig. 1E is presented does not seem to support this idea. How was day vs night total mating frequency calculated? Is ZT12 or ZT0 considered day or night?

What are the authors’ interpretation on clock mRNA cycling in WT Culex? Here clock mRNA expression display 2-3 peaks, which is inconsistent with prior literature (Rivas 2018), and male versus female expression patterns look different from each other.

Figure 2E & J: One reason for this could be because the animals are under LD condition, which can mask some of the effects circadian gene mutants can have. To measure circadian regulation, both mRNA abundance and mating frequency assays must be done under free running condition.

How was the timepoint of 20:00 chosen for extraction of CHC? For figure 4C&D, how many days post dsRNA injection was the CHC collected? It’d be helpful to provide representative CHC profiles for dsRNA injected groups.

Reviewer #3: (No Response)

**Conclusions**

-Are the conclusions supported by the data presented?

-Are the limitations of analysis clearly described?

-Do the authors discuss how these data can be helpful to advance our understanding of the topic under study?

-Is public health relevance addressed?

Reviewer #1: Yes.

Reviewer #2: Discussion section summarizes main findings but lacks in scope. Authors should utilize this section to provide more insight.

Line 443-444: Needs clarification. Onset of daylight is ZT0/24. ZT3 is considered 3 hrs after daylight onset. Mating peak seems to be at ZT12, which at the transition of light/dark, not 3 hrs before.

Line 456-458: Needs clarification.

Do the authors have any thoughts on the sex differences of desat knockdown effects on mating? Are the affected CHC more specific to males? Or could desat affect biosynthesis of CHC differently in each sex?

Could the authors comment on what the implications of their findings may be in terms of vector control?

Reviewer #3: (No Response)

**Editorial and Data Presentation Modifications?**

Reviewer #1: NA

Reviewer #2: Use of red and green in the same graph is not recommended in order to keep graphs accessible to colorblind readers.

Incorrect figure and table references are found throughout. For example, reference to Figure S2 should be S1 in line 347 & 349; Figure S3 should be S2 in line 392; Table S3 should be Table S4 in line 176.

Line 147: “bodys” should be “bodies”

Line 213: I think the authors mean “3 hrs after perfuming”, rather than 3 hrs duration of perfuming.

Formatting errors throughout: missing space after comma, between sentences, or before parenthesis. 

Line 529: Perhaps the authors mean “that entrains to external light”?

Reviewer #3: (No Response)

**Summary and General Comments**

Reviewer #1: In the present manuscript, the authors analyzed the circadian nature of mating rhythms in two mosquito species of epidemiological importance: Aedes albopictus and Culex quinquefasciatus. Overall, the manuscript is well written and the logic flow between experiments is well articulated. 

I have only 2 main concerns and a series of minor points and questions. 

Main concerns:

Statistics: Instead of a chi-square (or Fisher) test, it would be more appropriate to use a test based on a binomial distribution. Ideally, a GLM with a binomial distribution of residuals or even better, a GLMM using the replicate number as a random effect. 

Similarly, instead of t-tests to compare mating rates, a similar strategy based on a GLM or GLMM, with a binomial distribution, would be better. 

Finally, an ANOVA also assumes a normal distribution of the data and, by definition, frequencies and percentages are not normally distributed. I strongly recommend that authors use a GLM with binomial distribution of residuals and a posthoc Tukey test for pairwise comparisons between treatments. 

The conclusions drawn from the reversed light cycle experiment are not clear. The authors state that these experiments demonstrate that the mating rhythm is entrainable, but since they were conducted under LD conditions, it could also simply be that a certain component of the rhythm is driven by external light conditions. An experiment where the light cycle is reversed for 3 days before testing mosquitoes under constant darkness conditions would be necessary to demonstrate that they have entrained to the reversed light cycle. Please clarify the interpretation of these results. 

Minor points:

Abstract 

L25: should “period” be plural? (since authors talk about the two transitions). 

L80: there is a missing “s” at D. simulans

L89: in “Culex quinquefasciatus is a nocturnal, harassing mosquito”, what do you mean by harassing? Males are harassing females? Or that it is an aggressively anthropophilic mosquito? 

L89: replace “coordinately-guided” with “coordinated”

L102: consider replacing “constructed” with “established”

L104: “ablation” should not be used here as this refers to RNAi. Instead, “the expression of the gene desat1 was inhibited” would be better.

L181: CHCs are extracted at 20:00pm. To what ZT time does this correspond?

L196: replace “sec” with “a second”

L213: “After perfuming 3 hours,” did you mean “Three hours after perfuming,”?

L239: shouldn’t ZT12 be ZT9? (i.e., 3 hours before lights off?)

L251: “These results suggest that the circadian rhythms of mating activity can be precisely entrained by light signals.” Don’t you need to expose mosquitoes to this reversed light cycle and then transition to DD to show that they are entrained? Maybe what we are seeing is simply driven by external cues (light).

L263: “ There were no significant difference in the total mating rates of Ae. albopictus and Cx. quinquefasciatus between day and night under DD conditions (Fig 1E). These results indicate that the mating rhythms of the two species are completely opposite with Ae. albopictus mating during the day and Cx. quinquefasciatus mating at night.”

This paragraph is a bit unclear and needs to be reworded. What specific comparisons were made?

L310: “but there was no significant difference in daytime mating rates (P >0.05).” please add the values here for clarity.

L444: again, in “onset of day-light (ZT3) and 3 hours before darkness (ZT12)” Shouldn’t it be ZT9?

L467: “tus” is missing from “quinquefascia”

L526: delete “in” from “that involve in the regulation”

Figure 1

Panels B-D: an ANOVA is used to compare frequencies while an ANOVA assume normal distribution. Using GLM with a binomial distribution and posthoc Tukey tests for pairwise comparisons would be better. 

Panel D: So, with the reversed LD: what are we seeing? That rhythms are driven by light? Entrainment would require 3 days in reversed LD and then DD to see that they have entrained. The interpretation of these results needs to be clarified in the MS.

E: Binomial tests would be more appropriate. 

Figure 2:

Panels D-E: So, although the difference between the scotophase of WT and clock knockouts seems to be different (but see my previous comment about statistics), the overall profile is similar between WT and mutants. Showing that mating rhythms are driven, to a large extent by light conditions. An interesting experiment would have been to test the knockout line under DD. Can authors comment on that in the manuscript? This would help readers identify the outstanding knowledge gaps. 

Panels I-J: Here the phase inversion is interesting, but again, running the experiment under DD would have allowed the authors to detangle the effect of the circadian clock from the external light input. In any case, there seems to be an interaction between these two factors in Culex mosquitoes, which would be good to discuss in the manuscript. 

Figure 3

Same comment here: additional DD experiments would have allowed the distinction between light and clock effects. Could authors add a comment in the discussion?

It is interesting that in males, the inversion of the curves in A and C leads to an increase in the mating rates but with a peak at the same times of day. Shouldn’t we expect to see a shift in the time of peak mating frequency? And a reduction of the mating rate during peak hours (since desat1 is less expressed)?

Figure 4

One could argue that showing the unmated bars is redundant in panels F and G since the insemination rates already include that information. Please consider removing the grey bars.

Reviewer #2: Liu et al. investigate an interesting topic of circadian regulation of mating activity in Aedes albopictus and Culex quinquefasciatus mosquitoes. This reviewer agrees with the general conclusion and find the study interesting. There are parts that require clarification and discussion section lacks insight in its current state. Few of the conclusions are not supported by their data and need to be clarified.

Reviewer #3: The authors have done a great job of presenting the regulation of circadian mating activity in Aedes albopictus and Culex quinquefasciatus by clock genes. This study is quite important as it adds to the current body of knowledge of mating behavior and circadian rhythm of mosquitoes, and also provide insights for the development of mosquito control programs. The paper was clearly written, and the methods were well described. I have no big concerns with this manuscript; however, I would like to provide some few suggestions that will strengthen the manuscript.

1. This current study is similar to a previous investigation by Wang et al., 2021 (Clock genes and environmental cues coordinate Anopheles pheromone synthesis, swarming, and mating). In the 2021 study, the role of the clock genes (period and timeless) and desat1 was evaluated in three Anopheles species. While a different clock gene was investigated in two different mosquito species in this study, the authors need to state clearly a significant research gap this study is trying to address in mosquito mating behavior. I do not think the justification provided in the introduction was strong enough.

2. Line 90-91, "...the mating activity of Ae. albopictus and Cx. quinquefasciatus has yet to be determined molecularly" - is the use of "has" correct in this context?

3. There are some formatting issues in Lines 110, 114, 128 etc. The authors might want to do a thorough check through the whole manuscript.

4. Line 147 - "bodys" should be "bodies".

5. Lines 179 and 184 - The unit of measurement should be confirmed and corrected accordingly.

6. Line 210-211, "Every mosquito was applied ~1μL." - This is not clear.

7. Line 215-217 - Is there a reason for the difference in the mating duration for the two mosquito species? If yes, sufficient justification should be provided.

8. Line 275-276 (Figure 1 legend): No description of the gray portion of the bar.

9. Line 303 - "Circadian" should be starting with a lowercase "c".

10. Line 391 - compared should replace "comparted".

PLOS authors have the option to publish the peer review history of their article (what does this mean?). If published, this will include your full peer review and any attached files.

Reviewer #1: No

Reviewer #2: No

Reviewer #3: No
---

## [Decision Letter · Decision Letter 1]

16 Nov 2022

Dear Professor Chen,

Thank you very much for submitting your manuscript "Clock genes regulate circadian mating activity in the vector mosquitoes, Aedes albopictus and Culex quinquefasciatus" for consideration at PLOS Neglected Tropical Diseases. As with all papers reviewed by the journal, your manuscript was reviewed by members of the editorial board and by several independent reviewers. The reviewers appreciated the attention to an important topic. Based on the reviews, we are likely to accept this manuscript for publication, providing that you modify the manuscript according to the review recommendations. 

The reviewer have indicated that the paper is improved, but there are still minor issues that need to be addressed.

Sincerely,

Joshua B. Benoit

Academic Editor

Alvaro Acosta-Serrano

Section Editor

The reviewer have indicated that the paper is improved, but there are still minor issues that need to be addressed.

Reviewer's Responses to Questions

**Key Review Criteria Required for Acceptance?**

**Methods**

-Are the objectives of the study clearly articulated with a clear testable hypothesis stated?

-Is the study design appropriate to address the stated objectives?

-Is the population clearly described and appropriate for the hypothesis being tested?

-Is the sample size sufficient to ensure adequate power to address the hypothesis being tested?

-Were correct statistical analysis used to support conclusions?

-Are there concerns about ethical or regulatory requirements being met?

Reviewer #1: See below

Reviewer #2: (No Response)

Reviewer #3: The methods used in this study are adequate for the hypothesis being tested and suitable for the study system.

**Results**

-Does the analysis presented match the analysis plan?

-Are the results clearly and completely presented?

-Are the figures (Tables, Images) of sufficient quality for clarity?

Reviewer #1: See below

Reviewer #2: (No Response)

Reviewer #3: Results were properly presented and described.

**Conclusions**

-Are the conclusions supported by the data presented?

-Are the limitations of analysis clearly described?

-Do the authors discuss how these data can be helpful to advance our understanding of the topic under study?

-Is public health relevance addressed?

Reviewer #1: See below

Reviewer #2: (No Response)

Reviewer #3: Important conclusions and inferences made in this study are consistent with the data presented. Public health importance of this study was properly addressed.

**Editorial and Data Presentation Modifications?**

Reviewer #1: none

Reviewer #2: (No Response)

Reviewer #3: The authors might need to modify the sentence on lines 262-263 to read as follows: "Mosquitoes which received no solvent (blank group) and those that only got the n-hexane application (positive control) were included for comparisons".

**Summary and General Comments**

Reviewer #1: Overall, the authors have addressed the comments of the reviewers and added experimental details to the manuscript. Concerns about the statistical approach employed in the study have been addressed by using GLMs based on binomial distributions.

However, in the first part of the results section, the authors use wording suggesting that the GLM proves “distinct rhythmicity“. Rhythmicity needs to be tested by other statistical means. That being said, the authors could change the wording to “....activities were significantly influenced by the time of day”. Or something to this effect. 

L291: if measured under LD conditions, these are not “circadian” mating activities, but diel rhythms since they result from a combination of endogenous control and exogenous drive.

Reviewer #2: I thank the reviewers for their effort in addressing questions and concerns in the initial review. Two major concerns remain: (1) Results presented in Figures 2 and 3 still cannot conclude circadian effects as they were performed under LD. This is consistent with Reviewer #1’s initial concern. (2) Clk mutant mosquitoes were not properly outcrossed. Doing so is standard and necessary to rule out any genetic background effects, especially for behavioral tests. Both of these points should be clearly and properly addressed in the final version of this manuscript.

Overall, this manuscript has been largely improved with interesting and important findings. With the two major points listed above addressed properly, I would recommend this manuscript for publication.

Reviewer #3: I appreciate the authors for the clarifications provided to the concerns raised and the clear responses to the reviewers' comments. The study is important as it provides evidence to the role of the circadian clock in the mating behavior of Aedes albopictus and Culex quinquefasciatus. Conclusions derived from this study are relevant for devising novel mosquito control strategies. This study will consequently expand knowledge in the field of circadian biology and integrated pest management which is important not only to scientists in these fields, but also the general public. Therefore, I recommend it for publication.

PLOS authors have the option to publish the peer review history of their article (what does this mean?). If published, this will include your full peer review and any attached files.

Reviewer #1: No

Reviewer #2: No

Reviewer #3: Yes: Oluwaseun M. Ajayi

Figure Files:

Data Requirements:

Reproducibility:

References

---

## [Editor Report · Decision Letter 2]

20 Nov 2022

Dear Professor Chen,

We are pleased to inform you that your manuscript 'Clock genes regulate mating activity rhythms in the vector mosquitoes, Aedes albopictus and Culex quinquefasciatus' has been provisionally accepted for publication in PLOS Neglected Tropical Diseases.

Best regards,

Joshua B. Benoit

Academic Editor

Alvaro Acosta-Serrano

Section Editor

---

## [Editor Report · Acceptance letter]

28 Nov 2022

Dear Professor Chen,

We are delighted to inform you that your manuscript, "*Clock* genes regulate mating activity rhythms in the vector mosquitoes, *Aedes albopictus* and *Culex quinquefasciatus*," has been formally accepted for publication in PLOS Neglected Tropical Diseases.

Best regards,

Shaden Kamhawi

co-Editor-in-Chief

Paul Brindley

co-Editor-in-Chief
